# Secure Virtual Network Embedding Algorithms for a Software-Defined Network Considering Differences in Resource Value

**Ling Shen \*** , **Muqing Wu and Min Zhao**

School of Information and Communication Engineering, Beijing University of Posts and Telecommunications, Beijing 100876, China; wumuqing@bupt.edu.cn (M.W.); zhaomin@bupt.edu.cn (M.Z.)
\* Correspondence: lingshen@bupt.edu.cn

**Abstract:** Software-defined networking (SDN) and network virtualization (NV) are key technologies for future networks, which allow telecommunication service providers (TSPs) to share network resources with users in a flexible manner. Since TSPs have limited virtualized network resources, it is critical to develop effective virtual network embedding (VNE) algorithms for an SDN network to improve resource utilization. However, most existing VNE algorithms ignore the security issues of SDN networks, which may be subject to malicious attacks due to their openness feature. Therefore, it is necessary to develop secure VNE (SVNE) for SDN networks. In this paper, we researched the relationship between resource value and node security-level, and we found that there are differences in the resource value of different nodes. Based on this analysis, we define the evaluation indicators considering differences in resource value for the SVNE problem. Then, we present a mixed-integer linear program (MILP) model to minimize the cost of SVNE. As the formulated optimization problem cannot be solved conveniently, we design two node-ranking approaches to rank physical and virtual nodes, respectively, and we propose two novel SVNE algorithms based on the node ranking approaches. Finally, simulation results reveal that our proposed algorithm is superior to other typical algorithms.

**Keywords:** network virtualization; software-defined networking; secure virtual network embedding; security cost; security level

## 1. Introduction

Network virtualization (NV) is a key technology for future networks, which allows telecommunication service providers (TSPs) to share their virtualized network resources with their users in a flexible manner [1]. It is difficult for traditional network architectures to deploy NV technology due to their rigid structures and complicated service management mechanisms. Software-defined network (SDN) is a new network architecture, which separates the control plane and the data plane of network elements. NV decouples software-based virtual networks from hardware-based physical networks, so it is much easier to be implemented in SDN [2]. The combination of SDN and NV technology is considered as an effective way to overcome current network ossification and to promote future network innovation. Therefore, SDN-based network virtualization architecture has become a new research topic for scholars. Figure 1 shows an SDN-based NV architecture, which is proposed by Chai et al. [2].

The VN enables multiple tenants to share the same physical network that can create security vulnerabilities. NV has a positive effect on availability, but it has threatening security challenges related to confidentiality, integrity, and authenticity. Virtual networks can be created, deleted, and moved around a network easily, hence, tracking a malicious virtual network would be much more complex [3]. For example, a hypervisor may be hijacked; the SDN network hypervisors TeaVisor, DFVisor, Vertigo, TALON, and Sincon, fail

in terms of security. In addition, there are security problems in virtual resource allocation. Generally speaking, the security issue in virtual resource allocation can be classified into four types of security attacks: a physical element attacking a virtual element, a virtual element attacking a physical element, attacks among virtual elements, and attacks in the physical elements type [4,5].

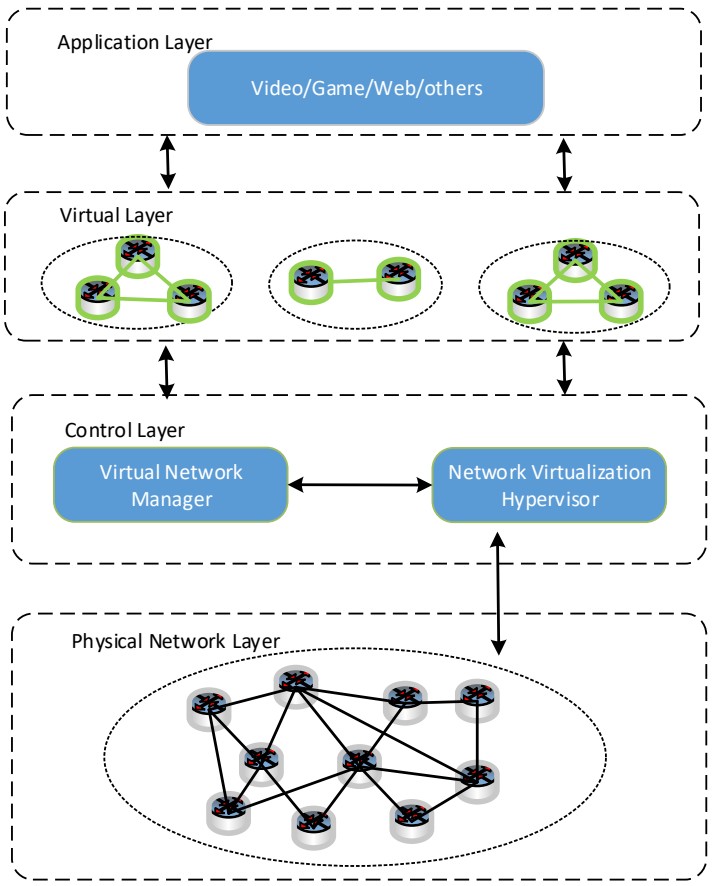

**Figure 1.** Virtualization-enabled SDN architecture.

Virtual resource allocation is also known as virtual network embedding (VNE), which is the key technology of NFV. An effective VNE algorithm can improve network resource utilization and accommodate more virtual network services. However, due to the complicated characteristics of the SDN substrate network, designing a VNE algorithm is difficult and challenging. Most of the existing VNE algorithms are researched and based on a traditional network, and they do not take into account the features of openness and programmability. Therefore, it is of great significance to research secure VNE (SVNE) in SDN-enabled networks. The core of the SVNE problem is to allocate network resources that meet the conditional restrictions to users according to business requirements. For example, the security requirements of online payment services are relatively high, and TSPs need to allocate the resources with higher security-levels to ensure the privacy of services. In general, the cost of the network resources is related to its security-level: the higher the security-level of the network resource, the higher its cost. In addition, the resource security-level determines the business scope of resource applications, which affects revenue. Resources with a high security-level are abstracted from high-security physical networks, and they are suitable for more business types than resources with lower security levels. Hence, there are differences in the value of resources at different security-levels. The resources with a high security-level have a higher value than a resource with a low security-level.

In this paper, we focus on the security issue in virtual resource allocation, and we research the relationship between the resource value and the security-level and then define the evaluation indicators that take into account the difference in the resource value. Then, we present a mixed-integer linear program (MILP) model with the goal of minimizing the SVNE cost since the formulated optimization problem is a complicated optimization problem, which cannot be solved conveniently. Therefore, we design two node-ranking SVNE algorithms, which is a two-stage algorithm called the T-SVNE algorithm and the node-link hybrid embedding algorithm, which is called H-SVNE algorithm. Overall, the major contributions of this paper are summarized as follows:

(1) We research the relationship between the resource value and its security-level, and we present a MILP formulation that takes into account differences in the resource value to minimize the embedding cost for the SVNE problem.

(2) Two SVNE algorithms are proposed based on node-ranking approaches to reduce the costs of VNE. The node-ranking approaches comprehensively considers the network topology, resources, and security-level attributes.

(3) Extensive simulation experiments are implemented to validate the performance of the proposed algorithms. Simulation results show that our proposed algorithms outperform selected typical algorithms.

The reminder of this paper is organized as follows. Section 2 summarizes the related work. Section 3 presents the network model, evaluation indicators, and the problem statement of SVNE. Section 4 introduces our proposed algorithms. Section 5 implements simulation work. Finally, we conclude this paper.

## 2. Related Work

### 2.1. SVNE Algorithms

Zhiming et al. [4] believe that network virtualization may cause information leakage via covert channels between virtual nodes coexisting on the same substrate node. This paper is the first to consider risk-tolerant coexistence in VNE. They propose a SVNE scheme to mitigate the risk of covert channel attacks. Boutigny et al. [6] ease the emergence of trusted brokers in between tenants and InPs for network virtualization. In addition, they present a VNE solution in a multi-provider context, as well as a use case demonstrating its feasibility. Zhang et al. [7] propose a security-aware VNE algorithm based on reinforcement learning. They add security requirement level constraints for each virtual node and a security level constraint for each substrate node. Virtual nodes can only be embedded on substrate nodes that are not lower than the level of security requirements. In [8], they propose a VNE algorithm with computing, storage resources, and security constraints to ensure the rationality and the security of resource allocation in ICPSs.

Cao et al. [5] attempt to tackle the security issues in 5G HetNets virtual resource allocation. The article starts from modeling the major security attacks for virtual resource allocation through comprehensive discussion of the typical types of security attacks. Following the attack model, a novel secure framework based on the emerging reinforcement learning approach is presented. Cao et al. [9] discuss the typical security risks in NFV-enabled networks. Then, they propose a secure framework. The goal of the framework is to ensure a secure network function deployment and resource allocation in NFV-enabled networks. In the framework, virtual network functions (VNFs), having high security probabilities, are usually preferred.

Chai et al. [2] research the VNE problem in SDN, where the substrate SDN switches and links may be subject to malicious attacks. They first propose a hierarchical virtualization-enabled SDN architecture and then formulate the VNE problem of SDN as a multi-objective optimization problem, which jointly minimizes network load and maximizes embedding reliability. In addition, the authors propose a virtual node embedding sub-algorithm and a virtual link embedding sub-algorithm to determine the locally optimal solution to the two sub-problems. Cao et al. [10] research the VNE for secure SDN-enabled networks. They first present a hierarchical virtualization architecture for SDN-enabled networks. Then,

the security model for different malicious attacks is presented. Finally, a novel SVNE framework is proposed for supporting SDN-enabled networks. In [11], they research the SVNE for SDN network, using blockchain technology. It is the first time that the SVNE algorithm is proposed for an SDN network. This paper focuses on considering the virtual network request (VNR) having the line topology, which is adopted to transmit data and directly share information.

### 2.2. Brief Summary

Most SVNE algorithms are developed for traditional network architectures, and they are not suitable for the SDN network environment. There are only a few articles that study the VNE problem in the SDN environment, where the substrate SDN switches may be maliciously attacked. In [2], they first propose a hierarchical virtualization-enabled SDN architecture based on which the VNE strategy can be designed; then, they formulate the VNE problem as a multi-objective optimization problem which jointly minimizes network load and maximizes embedding reliability. In [10], authors propose a hierarchical virtualization architecture for SDN networks; then they present the security model for different malicious attacks; finally, they propose a novel secure SVNE framework. In [11], authors research the SVNE for an SDN network, using blockchain technology. However, no scholars consider differences in the resource value. Therefore, we research the relationship between security and resource value, and define the evaluation metric of SVNE.

### 3. System Model and Evaluation Indicators

### 3.1. Network Model

We model the substrate node as a weighted undirected graph $G_s = (N_s, L_s, C_n, C_l)$ [12–15], where $N_s$ denotes the set of physical nodes and $L_s$ denotes the set of physical links. $C_n$ and $C_l$ denote the attributes of the underlying nodes and the physical links, respectively. For the nodes, the attributes include the CPU, ternary content addressable memory (TCAM) capacity, and security-level. The security-level is determined by the possibility of being attacked, which originates from historical records of these SDN network elements [8–11] that refer to the security model of virtual resource allocation in reference [5]. The higher the security-level of the node, the lower the probability of the node being attacked. For physical links, we only consider the bandwidth attribute, and we do not consider the possibility of the substrate link being attacked in this paper. It is owing to the fact that two end nodes of the attacked link can be regarded as two attacked nodes [11]. Therefore, the link attack can easily be transformed into a node attack. A node attack is a fundamental attack [11]. The lower part of Figure 2 shows an SDN-enabled substrate network. The numbers next to the nodes represent available CPU resources, TCAM resources, and the security-level capability, respectively.

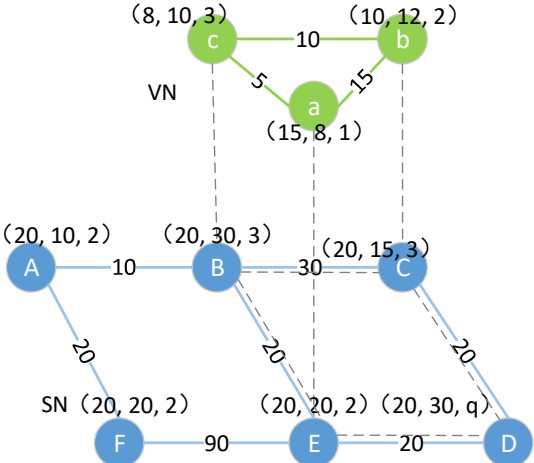

**Figure 2.** Example of VNE.

We also model the virtual network as a weighted undirected graph $G_v = (N_v, L_v, R_n, R_l)$; where $N_v$ denotes the set of virtual nodes; $L_v$ denotes the set of virtual links; and $R_n$ denotes the resource constraints of the virtual node, the virtual node computing capability requirement $CPU(n_v)$, the TCAM requirement $TCAM(n_v)$, the security-level requirement $SL(n_v)$; and $R_l$ denotes the virtual link bandwidth requirements $BW(l_v)$. A *VNR* can be expressed as a triad $VNR(G_v, t_a, t_d)$, where $t_a$ represents the arrival time of the *VNR*, and *td* represents the time when the virtual network leaves. The top of Figure 2 denotes a *VNR*, the numbers on the links denote the bandwidth requirements of links, and the three-dimensional arrays next to nodes represent the CPU, TCAM, and security-level requirements of the virtual nodes, respectively.

In order to facilitate the solution, a VNE problem can be abstracted into two parts, namely the node embedding and the link embedding parts. We express the VNE model as $G_V = (N_V, L_V) \rightarrow G_S = (N_S, L_S)$. As shown in Figure 2, the virtual node embedding result is: $\{a \rightarrow E, b \rightarrow C, c \rightarrow B\}$; and, the virtual link embedding result is: $\{(a,b) \rightarrow (E, D, C), (b,c) \rightarrow (C,B), (c,a) \rightarrow (B,E)\}$.

### 3.2. Evaluation Indicators

The main goal of VNE is to embed as many *VNRs* as possible to substrate networks. It is beneficial to increase the utilization of network resources and the revenue of TSPs. Most works use three metrics, i.e., average revenue, average revenue to cost ratio, and average acceptance ratio to measure the utilization efficiency of substrate network resources.

The value of network resources is related to the security-level; the higher the security-level of the network resources, the greater the cost. In addition, the resource security-level determines the type of service, which affects revenue. Therefore, we define the revenue of a *VNR* as follow:

$$R(G_v) = \sum_{n_v \in N_v} \{\alpha \cdot SL(n_v) \cdot (CPU(n_v) + TCAM(n_v))\} + \sum_{l_v \in L_v} BW(l_v) \tag{1}$$

where $\alpha$ represents the coefficient of the resource value.

The cost of a *VNR* represents the consumption of accepting a *VNR* that reflects the efficiency of the SVNE algorithm. Generally speaking, the lower the cost, the greater the economic benefit. The cost of a *VNR* can be calculated as follows:

$$C(G_v) = \sum_{n_v \in N_v, M(n_v) = n_s} \{\alpha \cdot SL(n_s)(CPU(n_v) + TCAM(n_v))\} + \sum_{l_v \in L_v} BW(l_v) \times hops(l_v) \tag{2}$$

We present the average revenue between $t_a$ and $t_b$ as follows:

$$R(G_v, t_a, t_b) = \frac{R(G_v, t_a, t_b)}{t_b - t_a} \tag{3}$$

We present the average revenue to cost ratio between $t_a$ and $t_b$ as follows:

$$R/C = \frac{R(G_v, t_a, t_b)}{C(G_v, t_a, t_b)} \tag{4}$$

The *VNR* acceptance ratio (AR) is an important metric in SVNE research, and it can be defined by:

$$AR(G_v, t_a, t_b) = \frac{VNR_{accept}}{VNR_{total}} \tag{5}$$

where $VN_{accept}$ represents the number of already accepted *VNRs* and $VN_{request}$ represents the number of arrived *VNRs*.

In addition, the average execution time for a *VNR* directly reflects the time complexity of the algorithm, which is also an important metric reflecting the efficiency of the VNE algorithms. We present the average execution time for a *VNR* between $t_a$ and $t_b$ as follows:

$$AET(G_v, t_a, t_b) = \frac{t_a - t_b}{VNR_{total}} \tag{6}$$

### 3.3. Problem Formulation of SVNE

In this subsection, we formulate the MILP model to solve the SVNE problem with the aim of minimizing the cost for accommodating the *VNR*. The main index notations throughout this paper are listed in Table 1.

**Table 1.** Notations.

| | |
|---|---|
| $G_s = (N_s, L_s)$ | Substrate network. |
| $G_v = (N_v, L_v)$ | Virtual network. |
| $i, j$ | Substrate nodes. |
| $u, v$ | Virtual nodes. |
| $SL(i)$ | The security-level of the physical node $i$. |
| $SL(u)$ | The security-level of the virtual node $u$. |
| $l_{ij}$ | Virtual link. |
| $l_{uv}$ | Virtual links. |
| $CPU(i)$ | Calculate ability of the node $i$. |
| $TCAM(i)$ | TCAM capacity of the node $i$. |
| $BW(l_s)$ | Link bandwidth of substrate link $l_s$. |
| $TCAM(u)$ | TCAM requirement. |
| $CPU(u)$ | Node calculate requirement of virtual node $u$. |
| $BW(l_{uv})$ | Link bandwidth requirement of the link $l_{uv}$. |
| $P_{st}$ | The physical path from node $s$ to $t$. |
| $\alpha$ | The coefficient of resource value. |
| $\beta$ | The weight coefficient. |

Variables:

$f_{ij}^{uv}$: a binary variable, its value is 1 if the substrate path $l_{ij}$ accommodates the virtual link $l_{uv}$; otherwise, the value is 0.

$x_s^u$: a binary variable, its value is 1 if the virtual node $u$ is embedded onto the substrate node $s$; otherwise, the value is 0.

The objectives of the optimization algorithm can be defined as follows:

$$Min \left( \sum_{(i,j) \in G_s} \sum_{(u,v) \in G_v} f_{ij}^{uv} \cdot BW(l_{uv}) + \sum_{n_v \in N_v, M(n_v) = n_s} \{ \alpha \cdot SL(n_s) \cdot (CPU(n_s) + TCAM(n_s)) \} \right) \tag{7}$$

Capacity constraints:

$$\forall u \in N_v, \forall i \in N_s, x_i^u \cdot CPU(u) \leq CPU(i) \tag{8}$$

$$\forall u \in N_v, \forall i \in N_s, x_i^u \cdot TCAM(u) \leq TCAM(i) \tag{9}$$

$$\forall l_{ij} \in L_s, \forall (u,v) \in L_v, f_{ij}^{uv} \cdot BW(l_{uv}) \leq BW(l_{ij}) \tag{10}$$

Security-level constraints:

$$\forall u \in N_v, \forall i \in N_s, x_i^u \cdot SL(u) \leq SL(i) \tag{11}$$

Connectivity constraints:

$$\forall i \in N_s, \forall l_{uv} \in L_v, \sum_{(i,j) \in L_s} f_{ij}^{uv} - \sum_{(j,i) \in L_s} f_{ji}^{uv} = \begin{cases} 1 & if \ x_i^u = 1 \\ -1 & if \ x_i^v = 1 \\ 0 & otherwise \end{cases} \tag{12}$$

Variable constraints:

$$\forall i \in N_s, \sum_{u \in N_v} x_i^u \leq 1 \tag{13}$$

$$\forall u \in N_v, \sum_{i \in N_s} x_i^u = 1 \tag{14}$$

$$\forall i \in N_s, \forall u \in N_v, x_i^u \in \{0,1\} \tag{15}$$

$$\forall l_{ij} \in L_s, \forall l_{uv} \in L_v, f_{ij}^{uv} \in \{0,1\} \tag{16}$$

Constraint (8) guarantees that the CPU capacity of substrate node i can satisfy the CPU requirements of virtual node $u$. Constraint (9) guarantees that the TCAM capability of substrate node $i$ can satisfy the TCAM requirements of virtual node u. Constraint (10) guarantees that the bandwidth capacity of substrate link $l_{ij}$ can satisfy the bandwidth requirement of virtual link $l_{uv}$. Constraint (11) guarantees that the security of substrate node i can satisfy the security requirements of virtual node *u*. The connectivity constraint in Constraint (12) is a flow conservation constraint. Constraint (13) guarantees that a substrate node can host at most one virtual node from the same *VNR*. Constraint (14) guarantees a virtual node can only be embedded onto one substrate node.

## 4. Proposed Solution

As we all know, the SVNE problem is a complex optimization problem, and the computation time cannot satisfy actual needs when the network size is large. Therefore, we propose two novel heuristic algorithms for the SVNE problem. In this section, we first introduce proposed node-ranking approaches, and then we propose two SVNE algorithms based on the node-ranking approaches. Finally, we analyze the complexity of the algorithms.

### 4.1. Node-Ranking Approach

(1) ***Virtual Node-Ranking Approach:*** The virtual nodes of the Connection-Bandwidth are defined according to Formula (17). The role of Formula (18) is to reorder virtual nodes with equal Connection-Bandwidths according to security-level. We select the node with the largest *NRV* as the first embedded node. The remaining virtual nodes are ranked according to Formulas (19) and (20). The $N_r$ represents the set of virtual nodes, which have been ranked.

$$CB(u) = \sum_{l_u \in nbr(u)} BW(l_u) \tag{17}$$

$$NRV(u) = \beta \cdot sl(u) + CB(u) \tag{18}$$

$$CB'(u) = \sum_{v \in N_r} BW(l_{uv}) \tag{19}$$

$$NRV'(u) = \beta \cdot sl(u) + CB'(u) \tag{20}$$

where $\beta$ represent the weight coefficient, and its value makes $\beta \cdot sl(u)$ less than 1. Its role is to reorder virtual nodes with equal Connection-Bandwidths according to security-level.

For ease of understanding, we take an example to illustrate virtual node-ranking process. We set the values of the parameter $\beta$ to 0.1. The VN in Figure 3 has four virtual nodes, their Connection-Bandwidth are 23, 30, 11, and 24, respectively, and their security-level are 1, 1, 3, and 2, respectively. Hence, we can obtain their node-ranking values *NRV* as: 23.1, 30.1, 11.3, and 24.2, respectively. Thus, virtual node b is selected as the first embedding node, and it is put into $N_r$. We can calculate Connection-Bandwidth of a, c, and d, which are 15, 0, and 15 based on Formula 18; and, they obtain their node-ranking values *NRV'* as: 15.1, 0.3, and 15.2. Thus, virtual node d is selected as the second embedding node, and it is put into $N_r$. At this time, $N_r$ includes nodes b and d. We can get the Connection-Bandwidths of a and c to be 18 and 6, and the node-ranking values *NRV* as: 18.1 and 6.3. Therefore, virtual node a is selected as the third embedding node, and d is the last mapping node.

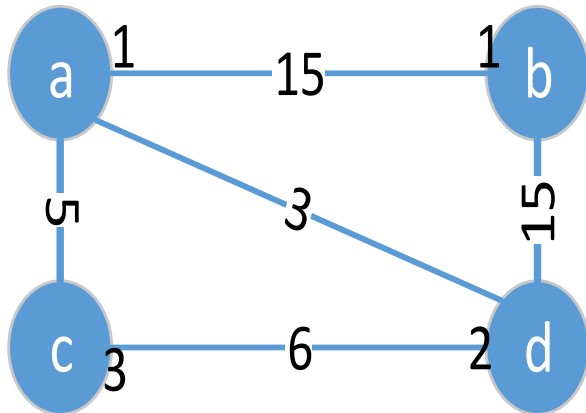

**Figure 3.** Example of virtual node ranking.

(2) ***Physical Node-Ranking Approach:*** Physical nodes are ranked according to the value of Bandwidth-Distance, which is our proposed node-ranking approach in [16]. When we embed the virtual node *u*, we can get the Bandwidth-Distance of the physical node *n* according to Formula (21).

$$BD_n = \sum_{v \in N_v} Dhop(n,m) * BW(l_{uv}), \ M(v) = m \tag{21}$$

where $Dhop(n,m)$ denotes the hops of the shortest path, $BW(l_{uv})$ denotes the connection bandwidth of virtual link $l_{uv}$, *v* denotes the other nodes belong to the same VN as *u*, and m denotes the physical node, which hosts virtual node *v*. The smaller the value of $BD_n$, the more bandwidth resources may be saved in the virtual link embedding phase.

In order to reduce cost, virtual nodes should be embedded to physical nodes that meet the conditions with lower security-levels. Therefore, physical nodes are sorted according to Formula (22).

$$NRP(n) = BD(n) + \beta \cdot SL(n) \tag{22}$$

We also take an example to illustrate the physical node-ranking approach. We assume that there are four physical nodes *A*, *B*, *C*, and *D*, and their security-levels are 4, 3, 2, and 3, respectively. When we embed virtual node *u*, the *BDs* values are 30, 20, 20, and 30, respectively. Hence, the value of *NRP* can be obtained as 30.4, 20.3, 20.2, and 30.3 according to Formula (21). They are sorted in ascending order according to the value of *NRP*, and they obtain the sequence *C*, *B*, *D*, and *A*.

*4.2. SVNE Algorithms*

Algorithm 1 presents T-SVNE, which includes the node and the link embedding stages. We find the *Dhops* between all physical nodes in advance to reduce the running time. Virtual nodes are sorted according to the virtual node-ranking approach. To embed virtual node *u*, physical nodes are ranked according to physical node-ranking approach. If the physical node *i* can meet the requirements of the virtual node *u*, we embed the virtual node *u* onto the physical node *i*. The algorithm performs link embedding when all virtual nodes are successfully embedded. The *VNR* is embedded successfully if all virtual nodes and links are successfully embedded.

---

**Algorithm 1:** T-SVNE Algorithm

---

Input: $G_s$, $G_v$, *Dhop*.
Output: embedding solution.
1: Sort the virtual nodes based on the ranking approach;
2: **for all** virtual node $u$ **do**
3:      Sort physical nodes based on physical node-ranking approach;
4:      **for all** physical node $n$ **do**
5:        **if**   $CPU(u) \leq CPU(n)$, $TCAM(u) \leq TCAM(n)$ and $SL(u) \leq SL(n)$ **then**
6:          Embed virtual node $u$ to physical node $n$, $M(u) = n$;
7:          **break**
8:        **end if**
9:      **end for**
10: **end for**
11: Sort the virtual links based on the bandwidth requirements in descending order;
12: **for all** virtual links **do**
13:      Use Dijkstra algorithm [1,17,18] to find the shortest path;
14: **end for**

---

*4.3. H-SVNE Algorithm*

To improve the efficiency of link embedding, we propose the H-SVNE algorithm. The core idea of H-SVNE is to pre-map virtual node $u$ to each physical node, and select the physical node with the least bandwidth consumption as the destination node. In order to reduce the execution time, the physical nodes and links that do not meet resource conditions are deleted. The algorithm is presented in Algorithm 2, which is also designed based on the above node-ranking approaches. Lines 3–13 are for constructing the BD matrix, and we can get the BD of the physical node $n$ based on the BD matrix. We can get the *NRP* of physical nodes based on Formula (21), and then sort the physical nodes in ascending order based on the value of *NRP*.

---

**Algorithm 2:** H-SVNE Algorithm

---

Input: $G_s$, $G_v$.
Output: embedding solution.
1: Sort virtual nodes based on the ranking approach;
2: *for all* virtual node $u$ *do*
3:      Delete physical nodes do not meet resource and SL requirements;
4:      *for all* physical node $n$ that meet requirements *do*
5:        Initialize $BW'(l_s) \leftarrow BW(l_s)$;
6:        *for all* physical node m hosts v that has a link with u *do*
7:          Delete the physical links ($BW(l_{uv}) > BW'(l_s)$);
8:          Use Dijkstra algorithm to find the shortest path;
9:          if there is the shortest path then
10:            Calculate $BD_{mn}$, record information and update $BW'$(ls);
11            *end if*
12:        *end for*
13:      *end for*
14:      BD matrix element column addition;
15:      Sort physical nodes based on NRP;
16:      *if* u is embedded successfully *then*
17:      Update BW(ls) if u is embedded successfully;
18:      *else*
19:        *break*
20:      *end if*
21: *end for*

---

*4.4. Time Complexity Analysis*

The time complexity of virtual node-ranking is $O(|N_v|)$, physical node-ranking is $O(|N_s|^3|N_v|)$, and the Dijkstra algorithm is $O(|N_s|^2)$. The T-SVNE algorithm consists of two sections: the time complexity of node-embedding stage is $O(|N_s|^2|N_v|)$, and the time complexity of link-embedding stage is $O(|Ns|^2|Lv|)$. Therefore, the time complexities of T-SVNE is $O(|N_s|^2|L_v| + |N_s|^2|L_v|)$ and the time complexities of H-SVNE is $O(|N_s|^3|L_v||N_v|)$. It is easy to observe that the time complexity of the H-SVNE algorithm is much higher than the time complexity of the T-SVNE algorithm.

## 5. Performance Evaluation

This section first describes the simulation parameter settings and the compared algorithms. Then, we present the experimental results in the form of graphs, and we analyze the experimental results in detail.

*5.1. Simulation Settings and Compared Algorithms*

The substrate network topology and the virtual network topology used in the simulation are generated by the improved Salam network topology random generation algorithm. We refer to the parameter setting in reference [19] (slightly modify); the substrate network (single domain) parameters are summarized in Table 2, and the VN parameters are summarized in Table 3, respectively. In addition, parameter $\alpha$ is set to 1, and parameter $\beta$ is set to 0.1. We use MATLAB R2021b as the simulation software to implement simulation experiments.

**Table 2.** Substrate Parameters.

| Physical Network Generation Approach | Salam Method, BorderLenght = 1000, Alpha = $10^{10}$, Beta = 0.25 |
|---|---|
| Node Capacity | [80, 100], uniform distributed |
| TCAM | [80, 100], uniform distributed |
| Link Bandwidth | [50, 80], uniform distributed |
| Number of nodes | 100 |
| Security level | [0, 4], uniform distributed |

**Table 3.** Virtual Parameters.

| Virtual Network Generation Approach | Salam method, BorderLenght = 1000, Alpha = $10^{10}$, Beta = 20 |
|---|---|
| VNR Arrival Rate | 4 *VNRs* per 100 time units |
| Number of virtual nodes | An integer, distributed [2, 10] |
| Node Capacity Demand | [25, 30], uniform distributed |
| TCAM Demand | [25, 30], uniform distributed |
| Link Bandwidth Demand | [25, 30], uniform distributed |
| Security level Demand | [1, 3], uniform distributed |

Four algorithms make up the simulation part in total. Besides our proposed T-SVNE and H-SVNE algorithms, the remaining algorithms are RCR-VNE [20] and NRM-VNE [20]. These algorithms are typical VNE algorithms, and they are slightly modified to suit the experimental scenarios.

*5.2. Simulation Results*

The main simulation results are presented in this subsection. Algorithms are tested based on the above simulation environment. Figure 4 presents the average acceptance ratio as a function of time. Observed from Figure 4, the curves of all algorithms decay with the variation of time. This decay shows that there are no infinite substrate resources to receive more and more *VNRs*. Our proposed algorithms perform better than other algorithms; it is owing to the fact that our algorithms embed virtual nodes to the physical nodes with smaller

Bandwidth-Distance, which is beneficial for saving bandwidth resources. In addition, H-SVNE has a higher acceptance rate than T-SVNE; it is owing to that H-SVNE adopts the traversal strategy to find the physical nodes with the least bandwidth consumption as the destination nodes, while the T-SVNE algorithm adopts the *Dhop* calculated in advance to replace the number of hops in the shortest path. Therefore, the accuracy of *Dhop* is not high, which affects the quality of the VNE solution.

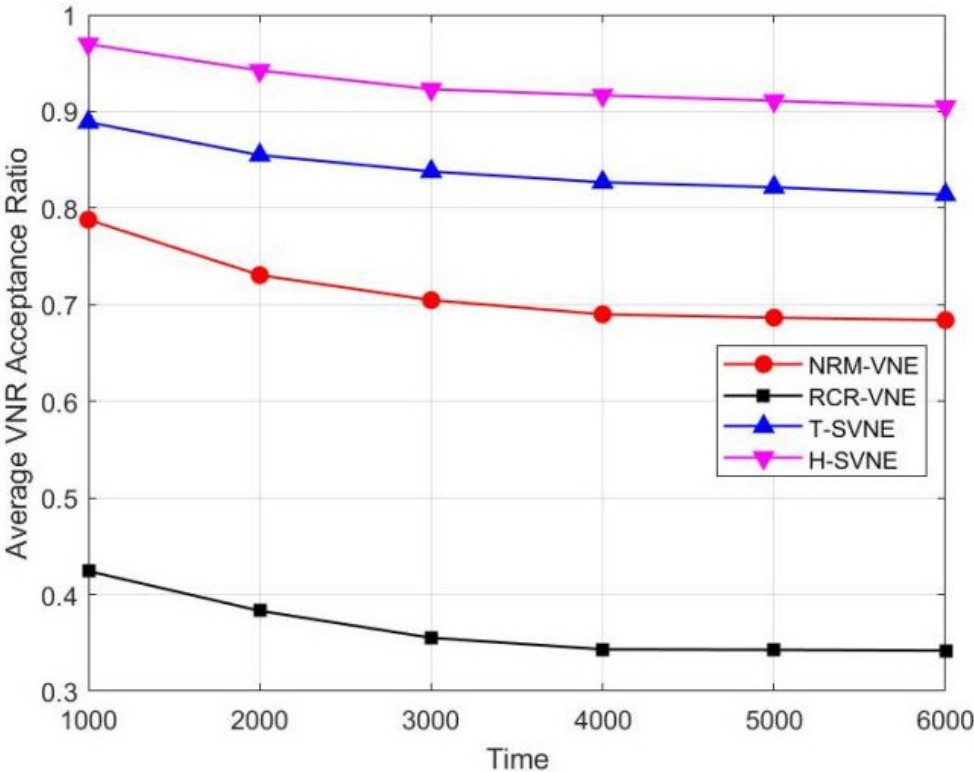

**Figure 4.** The average acceptance ratio.

Figure 5 plots the average embedding revenues of all selected algorithms. As observed in Figure 5, we can see that the overall trend of these algorithms is decreased with the running time, it is owing to the fact that the acceptance rate of all algorithms decreases over time. Our proposed algorithms earn more embedding revenues than the compared algorithms. This is due to two reasons: (1) our proposed algorithms accommodate more *VNRs* in the long run; (2) our proposed algorithms embed virtual nodes to physical nodes with a lower security-level. Moreover, the HSVNE algorithm is more profitable than the TSVNE algorithm, this is due to the fact that H-SVNE accepts more *VNRs* than T-SVNE algorithm.

Figure 6 illustrates the average revenue to cost ratio as a function of time. As observed in Figure 6, the average revenue to cost ratio of all algorithms decreases over time. It is owing to the fact that the bandwidth resources of the links are fragmented as the VNE progresses; and virtual nodes are embedded onto physical nodes with larger Bandwidth-Distance. Our proposed algorithms perform better than other algorithms. This is due to two reasons: (1) our algorithms embed virtual nodes to the physical nodes with smaller Bandwidth-Distance; (2) our proposed algorithms embed virtual nodes to physical nodes with lower security-level, which are beneficial for reducing embedding costs.

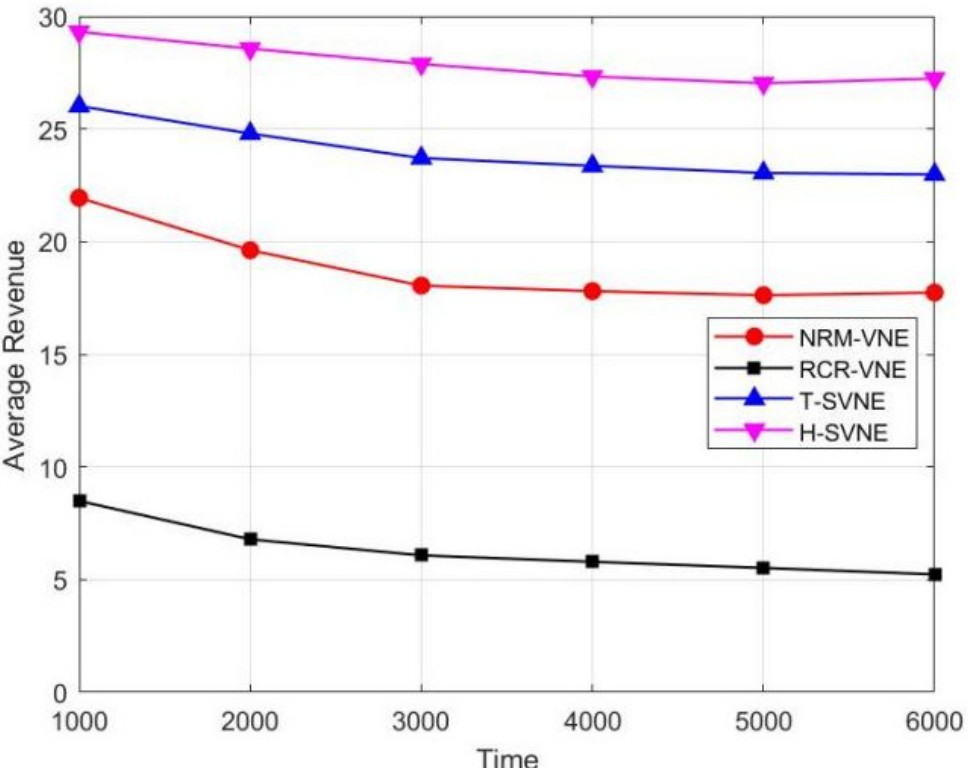

**Figure 5.** The average revenue.

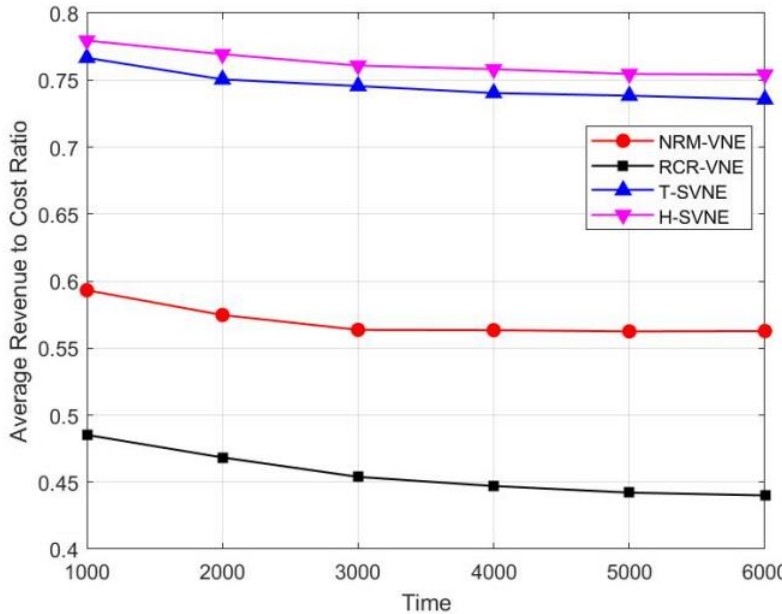

**Figure 6.** The average revenue to cost ratio.

Figure 7 shows the average execution time to process a *VNR*. From Figure 7, we can observe that the average execution time of the H-SVNE algorithm is much longer than the execution time of the T-SVNE algorithm. It is owing to the fact that the H-SVNE algorithm takes longer to traverse the shortest path of all physical nodes. Compared with T-SVNE, H-SVNE performs better in terms of average acceptance ratio, average revenue, and average revenue to cost ratio; however, H-SVNE takes more execution time to process a *VNR*. Hence, both algorithms have their own advantages and disadvantages.

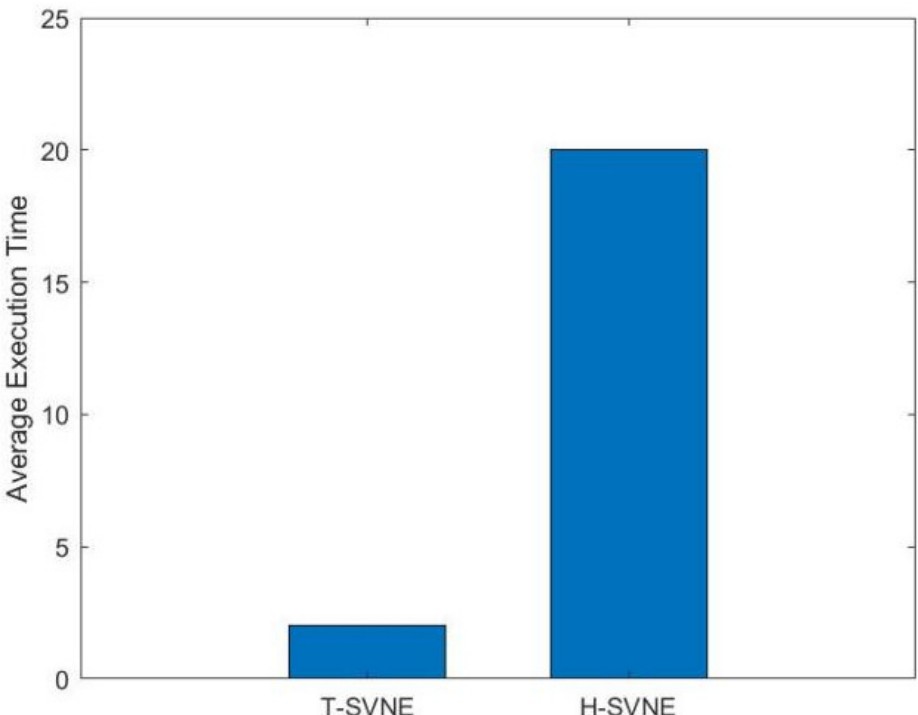

**Figure 7.** The average execution time.

## 6. Conclusions

In this paper, we research the relationship between resource value and security-level; and, we believe that the higher the security-level of the resource, the greater the value of the resource. Hence, we first define the evaluation indicators that take into account the difference in resource value and then present a MILP model with the goal of minimizing the SVNE cost. The MILP cannot be solved conveniently since the formulated optimization problem is a complicated optimization problem. Therefore, we propose two node-ranking strategies for sorting physical nodes and virtual nodes, respectively. Based on the node-ranking strategies, two novel SVNE algorithms are proposed: they are two-stage algorithms called the T-SVNE algorithm and the node-link hybrid embedding algorithm called the H-SVNE algorithm, respectively. Simulation results validate the effectiveness of our proposed algorithms. Moreover, the quick advances in artificial intelligence (AI) applications in social computing have led to an emerging and a promising study field known as artificial social intelligence (ASI) [21]. The virtualization of the intelligent network is the focus of future research.

**Author Contributions:** L.S. was responsible for the algorithm design, design experiments, data analysis, and writing the paper; M.Z. and M.W. checked the work. All authors have read and agreed to the published version of the manuscript.

**Funding:** This research was funded by the 111 project (NO.B17007), Director Funds of Beijing Key Laboratory of Network System Architecture and Convergence (NO.2017BKL-NSAC-ZJ-01) and the National Natural Science Foundation of China (NSFC) (NO.61872401).

**Conflicts of Interest:** The authors declare no conflict of interest.

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
