# Peer review of "Secure Virtual Network Embedding Algorithms for a Software-Defined Network Considering Differences in Resource Value"

_electronics, doi:10.3390/electronics11101662_

Round 1

Reviewer 1 Report

The paper proposes virtual network embedding algorithms for network virtualization based on software-defined networking. In particular, the authors suggest two kinds of algorithms, called T-SVNE and H-SVNE, incorporating both security requirements on network nodes and resource requirements. The suggested algorithms are evaluated through simulations.

The paper is relatively well-written and I have a few comments to improve its readability.

1. How is the security level gained in detail? The paper describes it as "by being attacked possibility originate from historical records of these SDN network elements." However, I observe examples of integer values (Fig. 2) and the evaluation settings of uniform distributed values between 0 and 4 or 1 and 3 (Tables 2 and 3). Is this a common approach when considering security? 

2. How can the various security concerns be incorporated into a single constant dimension?

3. I recommend that the authors consider summarizing the network virtualization schemes in SDN in order to enhance the importance of this paper. The SDN network hypervisors OVX, Libera, and CoVisor lack network embedding. They rely on static mappings between physical and virtual resources that cannot meet the resource or security needs of tenants. In addition, a number of approaches, such as TeaVisor, DFVisor, Vertigo, TALON, and Sincon, fail in terms of security.

4. Lastly, careful consideration of the entire paper for terminologies is required. For example, the abbreviation NV is used without being defined.

Author Response

Thank you for your comments and suggestions. Please see the attachment

Reviewer 2 Report

- you need to justify all your choices, or back your claim with relevant references. For example, in experiment, you claim 'parameter Alpha is set to 1, and parameter Beta is set to 0.1, These parameters are typical for VNR simulation.' you should explain why these values are typical for VNR simulation.
-since the Dijkstra algorithm is N square, why did you use it, there are many shortest path algorithms whose complexity is less than that.
-the following related work are missing:
[1]Yan, Zhongxia, et al. "Automatic virtual network embedding: A deep reinforcement learning approach with graph convolutional networks." IEEE Journal on Selected Areas in Communications 38.6 (2020): 1040-1057.
[2] Dehury, Chinmaya Kumar, and Prasan Kumar Sahoo. "DYVINE: Fitness-based dynamic virtual network embedding in cloud computing." IEEE Journal on Selected Areas in Communications 37.5 (2019): 1029-1045
[3] Dhelim et al. "IoT-enabled social relationships meet artificial social intelligence." IEEE Internet of Things Journal 8.24 (2021): 17817-17828.

Author Response

Thank you for your comments and suggestions. Please see the attachment.
